# Proton or Carbon Ion Therapy for Skull Base Chordoma: Rationale and First Analysis of a Mono-Institutional Experience

**DOI:** 10.3390/cancers15072093

**Published:** 2023-03-31

**Authors:** Slavisa Tubin, Piero Fossati, Ulrike Mock, Carola Lütgendorf-Caucig, Birgit Flechl, Maciej Pelak, Petra Georg, Christoph Fussl, Antonio Carlino, Markus Stock, Eugen Hug

**Affiliations:** 1MedAustron Center for Ion Therapy, Marie Curie Strasse 5, 2700 W. Neustadt, Austria; 2Klinische Abteilung für Strahlentherapie—Radioonkologie, Mitterweg 10, 3500 Krems an der Donau, Austria; 3Universitätsklinik für Radiotherapie und Radio-Onkologie der Paracelus Medizinischen Privatuniversität, Müllner Hauptstraße 48, 5020 Salzburg, Austria

**Keywords:** skull base, chordoma, proton therapy, carbon ion therapy, particle therapy

## Abstract

**Simple Summary:**

Oncologic gross total resection of skull base chordoma remains elusive in many patients. Dose-escalated radiotherapy, preferably by proton therapy, is frequently used. We present the early analysis of a mono-institutional experience using proton or carbon ion therapy for skull base chordomas. Our initial 4-year clinical outcomes suggest excellent local control. Large tumor volume was related with worse local tumor control, underlining the importance of maximum debulking of large lesions.

**Abstract:**

Background: Skull base chordomas are radio-resistant tumors that require high-dose, high-precision radiotherapy, as can be delivered by particle therapy (protons and carbon ions). We performed a first clinical outcome analysis of particle therapy based on the initial 4-years of operation. Methods: Between August 2017 and October 2021, 44 patients were treated with proton (89%) or carbon ion therapy (11%). Prior gross total resection had been performed in 21% of lesions, subtotal resection in 57%, biopsy in 12% and decompression in 10%. The average prescription dose was 75.2 Gy RBE in 37 fractions for protons and 66 Gy RBE in 22 fractions for carbon ions. Results: At a median follow-up of 34.3 months (range: 1–55), 2-, and 3-year actuarial local control rates were 95.5% and 90.9%, respectively. The 2-, and 3-year overall and progression-free survival rates were 97.7%, 93.2%, 95.5% and 90.9%, respectively. The tumor volume at the time of particle therapy was highly predictive of local failure (*p* < 0.01), and currently, there is 100% local control in patients with tumors < 49 cc. No grade ≥3 toxicities were observed. There was no significant difference in outcome or side effect profile seen for proton versus carbon ion therapy. Five patients (11.4%) experienced transient grade ≤2 radiation-induced brain changes. Conclusions: The first analysis suggests the safety and efficacy of proton and carbon ion therapy at our center. The excellent control of small to mid-size chordomas underlines the effectiveness of particle therapy and importance of upfront maximum debulking of large lesions.

## 1. Introduction

Chordomas are rare tumors that originate from notochordal remnants. Approximately one-third of lesions are located in the skull base [1]. Although chordomas are almost universally located in the bones, extra-axial chordoma sites have also been reported [2,3,4,5]. Chordomas account for less than 5% of all bone tumors [6] and for less than 0.2% of all intracranial neoplasms [7].

Skull base chordomas are characterized by slow growth and high local invasiveness, clinically presenting as clivus-destructing masses often exerting compression of the nearby brainstem, optic pathways and carotid arteries [8]. Differential diagnosis includes, but is not limited to, chondrosarcoma, myoepithelial carcinoma, meningioma or metastatic carcinoma. A specific marker for distinction between chordoma and other skull base tumors is brachyury, a nuclear transcription factor-member of the T-box gene family [9].

In approximately 5% of cases, distant metastases to lung, brain or bone may occur even years after the initial disease presentation [10,11]. Surgery represents the mainstay treatment for skull base chordomas. However, complete tumor resection resulting in negative surgical margins frequently cannot be accomplished due to the anatomical complexity of the skull base region and the highly infiltrative nature of this disease, with frequent involvement of the surrounding vascular and neurological critical structures [12,13]. Surgical resection with maximal tumor excision followed by high-dose postoperative radiotherapy represents the standard of care. According to the available literature data, the reported 5-year overall survival and local control rates following surgery and adjuvant radiotherapy range from 60% to 80% [14,15,16,17], and 69.6% to 75.8% [18,19,20,21,22], respectively. Since chordomas are radio-resistant tumors, eradication requires high-dose radiation ranging from at least 74 Gy up to 78 Gy (1.8–2 Gy per fraction). Since this radiation dose largely exceeds the tolerance of the nearby critical organs, it is frequently delivered by protons or carbon ions because of their physical characteristics, which provide a steeper dose gradient when compared with photon radiotherapy. 

The aim of this first analysis was to assess and report on the early mono-institutional clinical outcomes of proton and carbon ion therapy for skull base chordomas. Additionally, the potential predictive factors of local failure, as well as our institutional developing treatment decision algorithm for proton therapy and carbon ion therapy, were presented and discussed. 

## 2. Materials and Methods

### 2.1. Participant Characteristics

The present retrospective study included 44 patients with histologically proven skull base chordoma who were treated at the MedAustron Center for Ion Therapy between August 2017 and October 2021. The minimum required follow-up time-period was 6 months. The patients received particle therapy within the 2 months following surgery. All patients had a Karnofsky Performance status of 70% or higher. Chordomas were located either in the upper (68%) or lower (32%) clivus. A total of 41 patients (93%) presented with at least one symptom at time of consultation, and 61% of patients with symptomatic tumor-related cranial nerve deficits. Of these, 81% had developed high, 8% middle and 23% low cranial nerve deficits depending on local tumor involvement (see Table 1). A total of 10% of patients had multiple cranial neuropathies. The majority of patients underwent a subtotal tumor resection (57%). Gross total resection was accomplished in 21% of patients, and 12% of patients underwent biopsy only or decompression (10%). A macroscopic tumor was identifiable in 79% of patients on the planning CT/MRI. In total, 2/44 (5%) patients had a high-grade chordoma, 12/44 (27%) had a low-grade chordoma and in 30/44 (68%), the grade was not specified by the pathologist. Seven patients (16%) developed new symptoms following surgery, including dysgeusia, dysosmia, dysarthria, palpebral ptosis, tongue lateral deviation, paresthesia and symptoms related to cerebrospinal fluid leak. The main patients’ and disease characteristics are summarized in Table 1.

All procedures performed in the present study were in accordance with the ethical standards of the Helsinki Declaration. Written informed consent was obtained from all patients. The study was approved by the local Ethics Committee under registration number GS1-EK-4/824-2022.

### 2.2. Patient Simulation and Immobilization

CT and MRI scans were performed with 2 mm slice thickness, as per institutional practice, and were subsequently co-registered at the level of the skull base region. Patients were positioned and immobilized using a thermoplastic mask and customized vacuum cushion. The simulation CT was performed without contrast media, while the MRI was obtained with contrast media and in the treatment position. The MRI protocol consisted of T1ce, T2w (including FLAIR and MultiVane techniques) and DWI sequences. 

### 2.3. Volume Definition

The gross tumor volume (GTV) corresponded to any macroscopic tumor identified on the planning CT and MRI. The corresponding high-dose clinical target volume (CTV2) included the residual, post-operative GTV plus areas at high risk of microscopic disease. Low dose CTV (CTV1) included CTV2 plus areas of low to intermediate microscopic risk. As a general rule, the entire clivus and the prepontine cistern were included. CTV1 usually included the sella and bilateral cavernous sinuses for upper clival tumors. Laterally, CTV1 did not extend beyond an intact petro-clival junction. The retropharyngeal space was not routinely included in patients without evidence of extracranial tumor extension. A margin of 3 mm was added for the planning target volume (PTV). Critical Organs at Risk (OARs) such as optic pathways, brainstem, cochleae, temporal lobes, blood vessels of the Circle of Willis, pituitary gland, hypothalamus and hippocampus, were delineated on MR images and subsequently adapted on the co-registered simulation CT images. 

### 2.4. Treatment Planning

Treatments were prescribed using either protons or carbon ions and normalized to the respective median PTV dose. Carbon ion therapy was introduced at MedAustron in July 2019, approximately 2.5 years after patient treatments began with protons. Particle therapy plans were calculated using the pencil-beam scanning technique on the native simulation CT scans using Ray Station versions 8 and 11 with Pencil Beam (for carbon ions) and Monte Carlo (for protons) algorithm (Figure 1). The institutional algorithm for selection of protons versus carbon ions was primarily based on favoring carbon ions for patients with poorer prognostic features (see below). 

### 2.5. Dose Prescription

Treatment plans were calculated to deliver 74–78 Gy (RBE) in 37–39 consecutive fractions at 1.8–2 Gy (RBE) per fraction with protons, or 66 Gy (RBE) in 22 consecutive fractions (3 Gy (RBE) per fraction) with carbon ions to the high-dose PTV2. PTV1 was treated up to 54 Gy (RBE) with protons or 45 Gy (RBE) with carbon ions, respectively. Reported doses in Gy were RBE-weighted doses calculated from the physical dose using the local effect model for carbon ions (LEM-I with free parameters: α_γ_ = 0.1 Gy^−1^, β_γ_ = 0.05 Gy^−2^, D_t_ = 30 Gy, nuclear radius 5 μm) [23]. For protons, a fixed RBE of 1.1 was assumed. Thirty-nine patients (89%) were treated with proton beam therapy and five (11%) with carbon ion radiotherapy. 

### 2.6. Follow-Up

The RECIST-based [24] assessment of treatment response was performed every 3–6 months after treatment for the first 2 years by using CT and MRI, followed by repeated scans annually thereafter. Toxicity was evaluated using CTCAE Criteria [25]. 

### 2.7. Statistical Analysis

Stata V.16 was used for all analyses. Progression-free survival (TFS), and OS were assessed using the Kaplan–Meier method. Locally controlled patients were censored at the time of their last follow-up or death, whichever occurred first. OS was calculated from the initiation of proton therapy until death or loss to follow-up (censored data). Tumor control was defined as the lack of progression by clinical or radiological assessment. Any enlargement of the tumor on subsequent radiological studies was considered a local recurrence. Categorical data were summarized by total number and percentage and compared using the Chi-square test for sample sizes > 5 and Fisher’s exact test for sample sizes ≤ 5.

The possible influence of tumor volume on tumor recurrence was tested by logistic regression with progression as the dependent, and tumor volume as the independent variable. Furthermore, a non-linear model was fitted (Formula (1)). The inflection point of the sigmoid curve was interpreted as the threshold for an increased risk of recurrence. A *p*-value less than 0.05 was regarded as significant in all tests.
(1)$$Recurrence\;Risk=\frac{1}{1+e^{\left(b_{0}\times Tumor\;Volume+b_{1}\right)}}$$

Formula (1): Model describing recurrence risk as depending on tumor volume.

## 3. Results 

The median follow-up time was 34.3 months (range: 1–55). Actuarial *local control* rates at 2- and 3 years were 95.5% and 90.9%, respectively. There were four local recurrences (9%) occurring at ten, twenty-seven, thirty-four and thirty-seven months following treatment (Figure 2). In addition, one patient did not respond to treatment; the tumor continued to progress under the treatment and ultimately resulted in tumor-related death one month following completion of treatment. This patient was censored as a local failure. There was no significant difference between patients treated with protons versus carbon ions.

Overall survival at 2- and 3 years was 97.7% and 93.2%, respectively. Three deaths (6.8%) occurred due to disease progression (Figure 3). In two patients, the cause of death was local progression. The third patient died of distant disease despite local control. The deaths occurred at 1 month in the patient with tumor progression under treatment and at 26 months in the other two patients. 

Progression-free survival at 2- and 3 years was 95.5% and 90.9%, respectively. A total of 5 patients (11%) experienced local disease progression occurring at 1, 26, 39, 43, and 55 months and 2 patients were diagnosed with distant disease (Figure 4). 

Patterns of local failure in five patients were analyzed according to various tumor- and treatment-related parameters (Table 2). 

Tumor volume at time of particle therapy was found to be predictive of local failure (*p* < 0.01) (Figure 5, Table 3). The median volume of the five recurrent tumors was 110.6 cc (49–242), while of the thirty-nine controlled tumors was 31.7 cc (0–253). No recurrence was observed in tumors measuring < 49 cc.

No significant correlation was detected for other tumor- or treatment-related factors. 

Treatment-related side effects: The majority of patients (91%) developed mild to moderate acute side effects only—either as a single side effect or in combination. The most frequent side effects were grades 1 or 2 fatigue (52%), headache (27%), nausea (25%), mucositis (23%), erythema (18%), alopecia (18%), loss of appetite (16%), dysgeusia (14%), weight loss (14%), tinnitus (14%) and insomnia (7%). No patient developed acute or early-late Grade ≥ 3 toxicities, and no new onset of cranial nerve deficits was observed during the observation period. Concerning surgical complications, 3/44 (6.8%) patients presented with cerebrospinal fluid (CSF) leak. All three patients were treated with proton beam therapy, which did not aggravate that condition.

No incidence of high-grade brain necrosis was recorded. Six patients (14%) experienced transient grade ≤2 radiation-induced contrast enhancement (RICE) occurring in the medial segment of the temporal lobes in proximity to the target and within the high-dose region. These lesions appeared after an average time of 14 months (range: 7–23) following treatment. No patient required medical treatment and symptoms resolved spontaneously. MRI findings regressed spontaneously after an average time-period of 10 months. 

Disease, treatment and outcome characteristics of the five patients treated with carbon ions are summarized in Table 4. Critical OAR abutment or compression was present in four/five patients and in two patients carbon ions were selected for larger volume disease. No patient exhibited high-grade (≥Grade 3) acute or late side effects during the observation period. Local control was maintained in four/five patients; in one patient the tumor continued to progress under treatment, leading to death of the patient shortly thereafter.

## 4. Discussion

Our preliminary results concur with previously published data and confirm the safety and efficacy of particle therapy for skull base chordoma. Outside a randomized study, a comparison with other published data is problematic due to intrinsic issues common to all retrospective comparisons and an inadequate staging system. The only available tool for risk stratification is the Sekhar preoperative grading system [26], which has been shown to be predictive of outcome in irradiated patients [27]. However, despite its merits, it does not provide therapeutic decision guidance for the following questions: Will a second surgical debulking improve the prognosis or is a large residual tumor after initial surgery a predictor of negative prognosis that cannot be further improved upon by an additional resection? Is the bulk of the residual tumor an independent factor, or is it a surrogate of brainstem and optic chiasm compression and, therefore, inadequate dose coverage? Tumor volume at the time of particle therapy, together with optic apparatus and/or brainstem compression and/or tumor differentiation grade, is a well-known independent risk factor predicting local failure [28,29]. In our series, only residual tumor volume was predictive of outcome. “Debulking”, “Subtotal resection” or “Partial resection” are relative, qualitative and descriptive terms only. Accepting the absolute volume amount as a parameter will permit an estimate of whether additional surgery will improve the diagnosis based on reduced tumor volume.

Grade of histologic chordoma differentiation represents another factor negatively affecting prognosis. Among the three histological chordoma types, the classic or conventional and chondroid predict a more favorable long-term prognosis compared to the dedifferentiated chordoma type, which behaves biologically as a more aggressive tumor resulting in a 3-year overall survival rate of 60% compared to 90% for the first two histologic types [30,31,32,33]. In our series, the differentiation grade was not predictive of outcome—probably a result of the small sample size.

The decision algorithm for patient selection for either proton or carbon ion radiotherapy at our center is presently based on a combination of the size of macroscopic residual disease and proximity to or compression of critical structures (Figure 6). We classify residual tumors as small (<25 cc) medium (25–50 cc) or large (>50 cc). These thresholds are based on published data as well as our results.

For small (<25 cc) tumors without brainstem and chiasm compression, most published data for protons indicate an excellent local control, in excess of 90% at 5 years and 85% up to 10 years [14,18,19]. Proton therapy, with a standard fractionation of 1.8–2.0 Gy (RBE) dose per fraction and total prescribed dose levels of a minimum 72–76 Gy (RBE), is well-established with a relative low risk profile for long-term adverse events.

For large (>50 cc) residual disease, we initially discuss the feasibility of further surgical debulking with the referring neurosurgeon. However, in clinical practice there are various factors—including clinical or logistical considerations or patient’s refusal—may preclude maximal debulking. In such cases, we select carbon ion radiotherapy. 

In the Japanese NIRS (National Institute of Radiological Sciences) experience, a 16fraction schedule has been used with excellent results [34] based on a volume threshold of 34.7 cc, which was larger than the threshold reported by most proton series. This indicates the potential advantage of carbon ions for bulky disease. 

In the Japanese NIRS experience, a prescription dose of 60.8 Gy (RBE) in 16 fractions of 3.8 Gy (RBE) was used. The dose constraint for the brainstem was D0.1cc at less than 40 Gy (RBE) [35]. Due to the different RBE model used (mMKM vs. LEM), these doses would correspond in our center to a prescription of 68.8 Gy (RBE) and a constraint for the brainstem of 46 Gy (RBE) [36]. 

In the German HIT (Heidelberg Ion Therapy Center) experience, a prescription of 66 Gy (RBE) in fractions of 3 Gy (RBE) was employed with a dose constraint for the brainstem of 54 Gy (RBE) [21,37]. Due to the smaller difference between prescription and OAR constraint, we selected the German approach.

At the CNAO (National Center for Hadrontherapy) in Italy, carbon ions were used for more advanced cases than protons, with excellent results [38]. However, their outcome appeared inferior to that of protons for favorable cases. These results highlight the importance of patient selection.

The safety profile of carbon ions for skull base radiotherapy is less well established compared with proton therapy. We prefer to employ protons in patients with multiple co-morbidities and/or potentially decreased normal tissue tolerance. 

In case of medium (25–50 cc) residual disease, we evaluated each patient individually and, in selected cases, performed comparative proton and carbon ion treatment plans. 

In Table 5, we summarize the potential pros and cons of PBT vs CIRT based on published evidence. 

We believe that carbon ion radiotherapy for skull base chordoma holds the prospect of improving outcomes in poor-risk patients. 

In the future, we intend to explore treatment intensification for very unfavorable cases within prospective trials focusing on the following two aspects: 

(i.)Dose escalation with an inhomogeneous boost to the portion of GTV not abutting the brainstem or optic chiasm, and,(ii.)Escalation of the dose constraint to the brainstem without increased risk of brainstem toxicity. This could be achieved with LET (linear energy transfer) painting.

## 5. Conclusions

Our early clinical results are in accordance with published data from other Carbon-Ion centers and suggest the safety and efficacy of particle therapy for skull base chordoma.

The objective amount of residual tumor burden, as identified and measured on postoperative imaging following resection, emerges as an important prognostic variable.

We have developed a preliminary decision algorithm to select patients for either proton or carbon ion radiotherapy. This algorithm will be periodically re-evaluated pending larger patient accrual and re-analysis with longer follow-up.

## Figures and Tables

**Figure 1 cancers-15-02093-f001:**
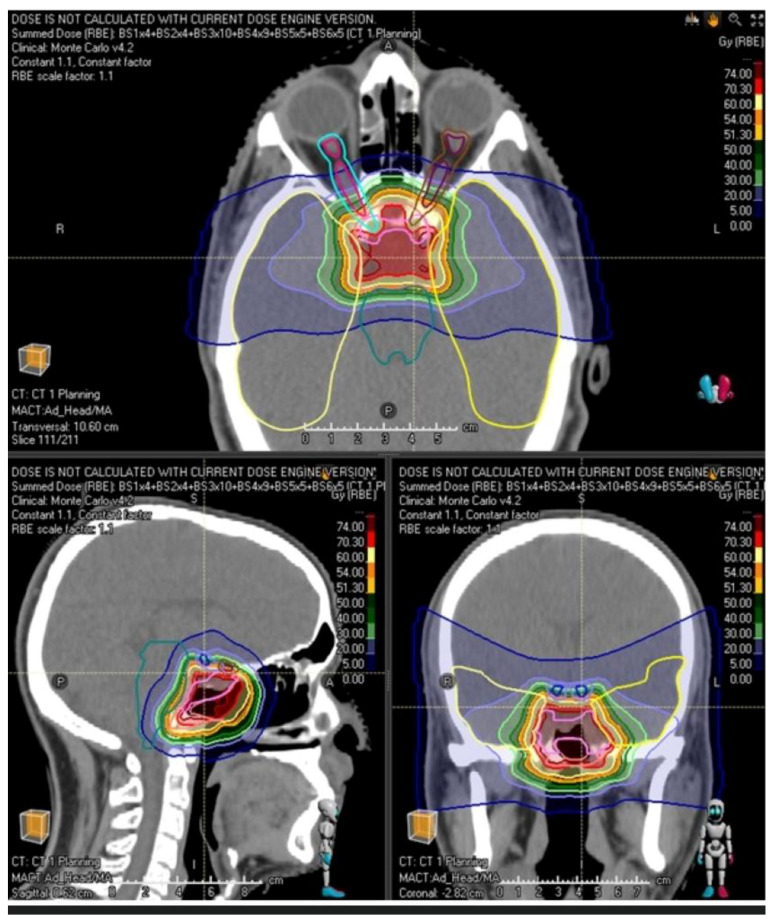
Proton beam therapy plan for skull base chordoma showing steep dose-gradient with sharp dose fall off in proximity of the optic pathways and brainstem.

**Figure 2 cancers-15-02093-f002:**
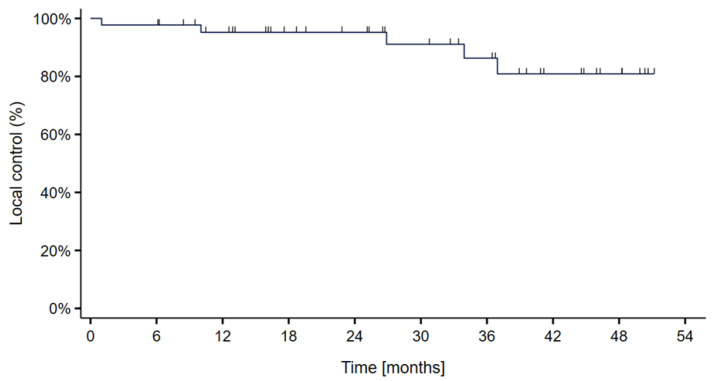
Kaplan–Meier curve showing the local control probability in 44 skull-base chordoma patients treated with particle therapy at MedAustron.

**Figure 3 cancers-15-02093-f003:**
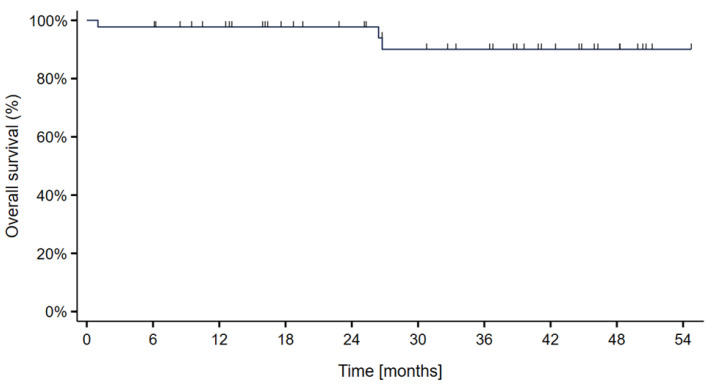
Kaplan–Meier curve showing the overall survival probability in 44 skull-base chordoma patients treated with particle therapy at MedAustron.

**Figure 4 cancers-15-02093-f004:**
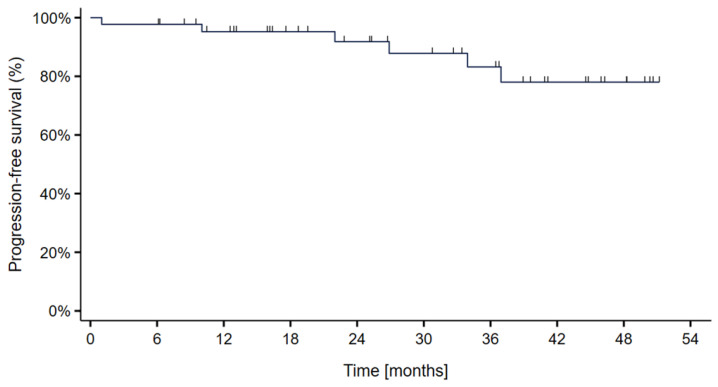
Kaplan–Meier curve showing the progression-free survival probability in 44 skull-base chordoma patients treated with particle therapy at MedAustron.

**Figure 5 cancers-15-02093-f005:**
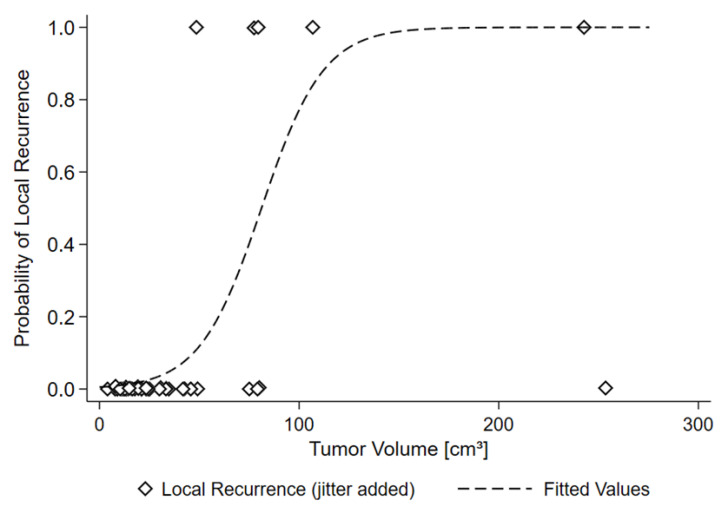
Relationship between tumor volume and local failure: tumor volume at time of particle therapy ≥ 50 cc was predictive of local failure. Graph depicts a non-linear model (sigmoid curve) for the relationship between tumor volume and local recurrence, *p* = 0.0097, *p*-value for logistic regression model for the association between volume and recurrence.

**Figure 6 cancers-15-02093-f006:**
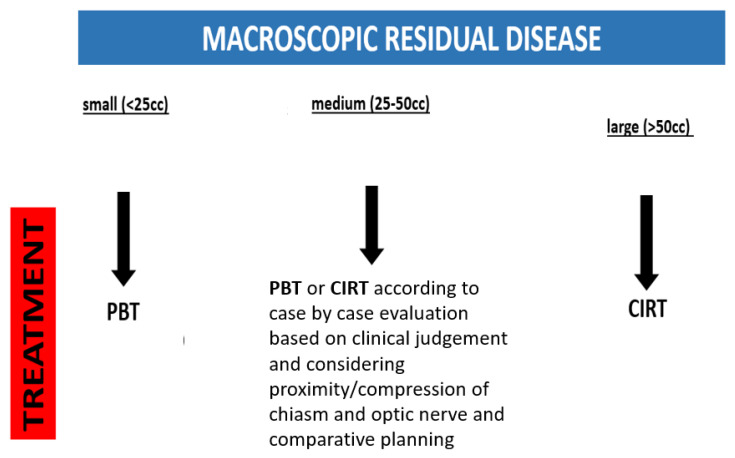
Present institutional treatment decision algorithm for PBT vs. CIRT based on the size of macroscopic residual disease, and involvement of critical structures PBT = proton beam therapy; CIRT = carbon ion radiotherapy.

**Table 1 cancers-15-02093-t001:** Patients and disease characteristics.

Sex:	Total Patients: 44
Male	24/55%
Female	20/45%
AGE:	Years
Average (range)	47 (19–87)
KPS: Median (range)	90% (60–100)
TUMOR SITE	
Upper clivus	30/68%
Lower clivus	14/32%
SURGERY:	
Subtotal tumor resection	25/57%
Gross total resection	9/21%
Biopsy	5/12%
Decompression	4/9%
TUMOR-RELATED SYMPTOMS:	41/93%
Cranial nerve deficit:	27/61%
- High: III, IV, VI (diplopia, ptosis)	22/50%
- Middle: V, VII (trigeminal neuralgia, facial paralysis/weakness)	3/7%
- Low: IX, X, XII (dysphagia and tongue lateral deviation)	8/18%
GROSS TUMOR VOLUME median (range) cm^3^	28.1 (1.4–218.9)
Brainstem or optic pathway compression/abutment	
- yes	25/57%
- no	19/43%
Vascular involvement (A. carotis/A. basilaris)	
- yes	2/5%
- no	42/95%
Intradural invasion	
- yes	3/7%
- no	41/93%

**Table 2 cancers-15-02093-t002:** Characteristics of five patients who experienced local failure following the particle therapy by MedAustron.

Local Failures	1	2	3	4	5
Tumor location	lower clivus	lower clivus	lower clivus	upper clivus	upper clivus
Tumor volume (ccm)	76	49	101	242	80
Vascular involvement	NO	NO	NO	NO	NO
Intradural invasion	NO	YES	NO	NO	NO
Brainstem/optic compression/abutment	YES	YES	NO	NO	YES
Type of particles	protons	protons	protons	protons	carbon
Radiation prescription dose (Gy RBE)	76	75.6	78.2	78	66
Dose to CTV1 95%	63.39	61.44	68.69	69.26	59.93
Dose to CTV1 98%	60.27	59.67	64.95	61.76	57.25
Dose to CTV2 95%	68.78	73.46	70.96	74.24	64.88
Dose to CTV2 98%	65.01	70.54	67.58	71.16	59.03
Time to recurrence (months)	10	34	27	37	1
Surgical resection	biopsy	decompression	subtotal	subtotal	biopsy
Radiation-induced brain tissue changes	NO	NO	NO	YES	NO
Alive at time of analysis: yes/no	NO	YES	YES	YES	NO
Follow up duration (months)	26	55	39	43	1

**Table 3 cancers-15-02093-t003:** Distribution of patients with tumor volumes </> 50 cm^3^ and local recurrences. Non-linear regression model reveals the above presented risks for local recurrence.

Tumor Volume	Local Recurrence	
	no	yes
<50 cm^3^	35 (89.7%)	1 (20.0%)
>50 cm^3^	4 (10.3%)	4 (80.0%)
	RISK OF LOCAL RECURRENCE	
50 cm^3^	11.8%	
81.1 cm^3^	50%	

**Table 4 cancers-15-02093-t004:** Disease and treatment characteristics of five patients treated with carbon ion therapy at MedAustron.

Patient	1	2	3	4	5
GTV pre-OP volume (cc)	48.64	79.63	8.31	24.64	80.46
Surgery	biopsy	debulking	debulking	debulking	biopsy
Site	lower clivus	lower clivus	upperclivus	lowerclivus	upper and lower clivus
CTV1 95% (Gy)	55.65	48.19	48.2	59.83	59.93
CTV1 98% (Gy)	51.84	45.65	46.3	58.08	57.25
CTV2 95% (Gy)	63.09	64.10	62.61	65.30	64.88
CTV2 95% (Gy)	59.65	62.41	58.05	64.39	59.03
Abutment or compression of optic structures and/or brainstem	YES	YES	YES	NO	YES
Toxicity	Mucositis G1, Fatigue G1	Tinnitus G1, Alopecia G2	Headache G2, Vertigo G2	Dysphagia G2, Mucositis G2	Nausea G2, Appetite loss G2
Radiation-induced brain tissue changes or brain necrosis	NO	NO	NO	NO	NO
Local control	YES	YES	YES	YES	NO
Follow up (months)	13	33	11	11	1

**Table 5 cancers-15-02093-t005:** Relative advantages and disadvantages of PBT and CIRT.

	Pros	Cons
PBT	Excellent result confirmed by multicentric series with large number of patients and long follow up	Larger spot size and less steep lateral penumbra possibly resulting in more undercoverage of target volumes in unfavorable cases
	Well-known toxicity profile and well-validated dose constraints	Significant dependency on residual tumor volume
CIRT	Smaller spot size and sharper penumbra potentially resulting in better target volume coverage also in unfavorable cases	More limited clinical experience available (fewer patients, shorter follow up)
	Potentially less dependent on residual tumor volume	Less well-established dose constraints for OARs

## Data Availability

Concerning the Data Availability Statements, the interested parties could directly contact the corresponding author.

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
