# Peer review of "Proton or Carbon Ion Therapy for Skull Base Chordoma: Rationale and First Analysis of a Mono-Institutional Experience"

_cancers, 2023, doi:10.3390/cancers15072093_

Round 1

Reviewer 1 Report (Previous Reviewer 2)

The authors present the revised version of their paper concerning a retrospective case series regarding the role of particle radiotherapy after surgical management of skull base chordomas; the principal drawback of this research is the unclearness of the aim.

As already discussed, the population included is too heterogenous concerning extent of tumor removal and radiotherapy protocol. Protocol application is not supported by evidence; it is undoubtedly true and intuitive that residual volume correlate with local recurrence.

Other several pitfalls have not been deeply discussed and should be revised by the authors before considering this manuscript for publication:

1.     Which is the time interval among surgery and RT? How about in case of surgical complications such as CSF leak or infections which can cause wound problems, aggravable by RT?

2.     Which is the performance status threshold for patients amenable of RT?

3.     Which was the histologic grade of the included patients?

Reviewer 2 Report (Previous Reviewer 3)

The authors answered satisfactorily to reviewer. 

This manuscript is a resubmission of an earlier submission. The following is a list of the peer review reports and author responses from that submission.

Round 1

Reviewer 1 Report

This is a relatively unique cohort of patients with skull base chordomas treated with proton and carbon ion therapy from a single center. The outcomes reported are excellent and are promising with regards to the techniques presented. The authors also identify a clear predilection for large tumors to recur, and present future trial ideas they plan to pursue to improve the outcomes of these patients. The results reported are a meaningful addition to the literature for this disease. The main limitation is the short follow up for outcomes and toxicity. The english and grammar could be improved and should be carefully reviewed, which would improve the manuscript. 

Author Response

- Comment: This is a relatively unique cohort of patients with skull base chordomas treated with proton and carbon ion therapy from a single center. The outcomes reported are excellent and are promising with regards to the techniques presented. The authors also identify a clear predilection for large tumors to recur, and present future trial ideas they plan to pursue to improve the outcomes of these patients. The results reported are a meaningful addition to the literature for this disease. The main limitation is the short follow up for outcomes and toxicity. The english and grammar could be improved and should be carefully reviewed, which would improve the manuscript. - Response: We have improved the English grammar with the support of a Native American medical-professional colleague. Thank you!

Reviewer 2 Report

In this paper, the authors present a retrospective case series regarding the role of particle radiotherapy after surgical management of skull base chordomas; the principal drawback of this research is the unclearness of the aim: indeed, neither a comparison among the two different particle radiotherapy protocols is performed nor suggestions for their application are provided. Reference no. 38 covers already most of these issues.

Furthermore, “Surgery represents a mainstay treat-64 ment of skull base chordomas whereby the amount of postoperative residual tumor tissue generally determines the prognosis”: this is undoubtedly true, but the population included is too heterogenous concerning extent of tumor removal: given the same radiotherapy protocol, it is obvious that gross total resection patients will survive longer than only decompressed ones. The key of the article are figure 5 and table 3, which deserve to be highlighted and discussed.

Moreover, the considerations discussed by the authors in the last sections are not supported or evident in the Results; which is the scientific relevance of authors’ recommendations?

Other several pitfalls should be discussed/revised by the authors before considering this manuscript for publication:

1.     Which is the time interval among surgery and RT? How about in case of surgical complications such as CSF leak or infections which can cause wound problems, aggravable by RT?

2.     Which is the performance status threshold for patients amenable of RT?

3.     How were the patients scheduled for a specific RT treatment?

4.     Which was the histologic grade of the included patients (i.e. how many classic/chindroid and how many dedifferentiated patients)?

Author Response

- Comment: English language and style are fine/minor spell check required. - Response: We have improved the English grammar. - Comment: In this paper, the authors present a retrospective case series regarding the role of particle radiotherapy after surgical management of skull base chordomas; the principal drawback of this research is the unclearness of the aim: indeed, neither a comparison among the two different particle radiotherapy protocols is performed nor suggestions for their application are provided. - Response: As you were aware of it, our 2 treatment groups are not yet comparable, due to the small sample size in the carbon-ion therapy group. Groups´ comparison will definitely be the aim of our next paper once the numbers of treated patients increase and become mature. However, we have been working on reviewer suggestions regarding the proton beam therapy and carbon-ion therapy applications, and those were included in DISCUSSION/CONCLUSION section. Additionally, the introductive part regarding the aims was also improved and adapted in this regard. - Comment: Furthermore, “Surgery represents a mainstay treatment of skull base chordomas whereby the amount of postoperative residual tumor tissue generally determines the prognosis”: this is undoubtedly true, but the population included is too heterogenous concerning extent of tumor removal: given the same radiotherapy protocol, it is obvious that gross total resection patients will survive longer than only decompressed ones. The key of the article are figure 5 and table 3, which deserve to be highlighted and discussed. Moreover, the considerations discussed by the authors in the last sections are not supported or evident in the Results; which is the scientific relevance of authors’ recommendations? - Response: According to the reviewer’s recommendation we have highlighted and discussed in greater detail the impact of macroscopic residual disease volume and our decision algorithm. We have rephrased the discussion to avoid possible confusion or misunderstanding. We respectfully disagree with the reviewer about the lack of value of describing the changes that we are implementing in our clinical practice based on our results and the published data, and the future development we plan to implement. We acknowledge that those changes and plans should not be presented as general recommendation to other centers but just as a description of our present strategy whose validity has to be validated. We have rephrased the discussion and conclusion section accordingly. - Comment: Other several pitfalls should be discussed/revised by the authors before considering this manuscript for publication: 1. Which is the time interval among surgery and RT? - Response: The time interval between the surgery and particle therapy was not exceeding 2 months. This was added to the Materials and Methods. - Comment: How about in case of surgical complications such as CSF leak or infections which can cause wound problems, aggravable by RT? - Response: 3/44 (6.8%) patients presented with CSF leak, no infections, all three treated with proton beam therapy; those surgical complications were not aggravated by particle therapy. 2. Which is the performance status threshold for patients amenable of RT? - Response: 70%.Added to the text. 3. How were the patients scheduled for a specific RT treatment? - Response: We started proton beam therapy in 2016 and carbon ion therapy in 2019, so that for those patients who started their therapy before 2019, carbon-ion therapy was not available. After it became available, we carefully started to introduce it in bulky tumors and tumors compressing the brainstem/chiasm. This empirical approach is confirmed by the preliminary results (at least in terms of need of more intense treatment, the effectiveness of carbon-ion therapy remains to be confirmed). Based on published data and on our preliminary results we have created a decision algorithm, including the new figure 6 to better explain our decision making process. 4. Which was the histologic grade of the included patients (i.e. how many classic/chindroid and how many dedifferentiated patients)? - Response: 2/44 (5%) dedifferentiated, 12/44 (27%) low-grade, and 30/44 (68%) not specified by the pathologist.

Reviewer 3 Report

The present study about the Proton and Carbon Ion Therapy to manage Skull Base Chordoma is well written and conducted; although the results sound and are clearly and honestly reported globally the study does not bring up neither new knowledge nor new management strategies to the existing body of literature. I believe that the authors should address the following problems before the study could be reconsidered for publication: 

1) The introduction section is too long (please shorten it).

2) In the methods section the authors state that inclusion criteria was a FU minimum of 6 months, on the other hand, they report patients with a FU between 1 and 55 months (please amend). 

3) We all know that talking about Chordoma, the definition of GTR is not easy (due to the intrinsic feature of such bony tumour as well as the involvement of important neurovascular structures) even if it represents a crucial feature for the overall and progression free survival; on the other hand in their study the authors report a GTR in 21%, a STR in 57% and a biopsy in 12% of patients respectively although vascular involvement (carotid and basilar arteries) and tumour intradural extension was present only in 5% and in 7% of patients respectively (please explain if possible the raison why further removal was not possible, please explain what by definition the authors mean for STR, actually what percentage of the tumour has been removed (i.e. ? 95%, ?? less, ??? between 96 and 99% ???? other).

4) The figure n°1 is not necessary (please remove it).

5) The volume definition section is too long (please shorten it). The dose prescription section is too long (please shorten it or alternatively synthetize it in a table).

6) The patient series treated by Carbon Ion Therapy is very small, do the authors really would like to report it? If yes it could of value the authors would add a table with all the pros, cons and indication of both proton and carbon ion therapies (please add).

7) The study could be greatly improved if the authors would add a clinical algorithm to guide clinicians in their routine practice when facing the radiotherapy of such patients.

Author Response

- Comment: The present study about the Proton and Carbon Ion Therapy to manage Skull Base Chordoma is well written and conducted; although the results sound and are clearly and honestly reported globally the study does not bring up neither new knowledge nor new management strategies to the existing body of literature. I believe that the authors should address the following problems before the study could be reconsidered for publication: 1. The introduction section is too long (please shorten it). - Response: this is done. 2. In the methods section the authors state that inclusion criteria was a FU minimum of 6 months, on the other hand, they report patients with a FU between 1 and 55 months (please amend). - Response: Let me put that in this way: we wanted as a minimum FU to be 6 months for all live patients that can be followed. Naturally, if someone will die earlier due to the PD, his/her FU will be shorter than 6 months, but since that is the outcome, must be included. And that actually happened. I guess we thought that the readers will understand where that contradictory number comes from, since in section OUTCOMES we described specifically that single case of fast progressing chordoma causing early death after only 1 month following the treatment. This is explained at the beginning of the “outcomes” section, please see the 3rd row. 3. We all know that talking about Chordoma, the definition of GTR is not easy (due to the intrinsic feature of such bony tumour as well as the involvement of important neurovascular structures) even if it represents a crucial feature for the overall and progression free survival; on the other hand in their study the authors report a GTR in 21%, a STR in 57% and a biopsy in 12% of patients respectively although vascular involvement (carotid and basilar arteries) and tumour intradural extension was present only in 5% and in 7% of patients respectively (please explain if possible the raison why further removal was not possible, please explain what by definition the authors mean for STR, actually what percentage of the tumour has been removed (i.e. ? 95%, ?? less, ??? between 96 and 99% ???? other). - Response: We have added a more detailed explanation about incomplete resection in our series. 4. The figure n°1 is not necessary (please remove it). - Response: With all due respect, I assume that the most of the potential readers of our paper and your journal are not the radiation oncologists, and especially not those who deal with the particle therapy. Having said that, we believe that the figure could show what words can´t tell to those readers who are not familiar with the particle therapy. The accent here is obviously on the specific ability of particles to maximally spare the nearby organs at risk. 5. The volume definition section is too long (please shorten it). - Response: this was done. 6. The dose prescription section is too long (please shorten it or alternatively synthetize it in a table). - Response: this was also done. 7. The patient series treated by Carbon Ion Therapy is very small, do the authors really would like to report it? If yes it could of value the authors would add a table with all the pros, cons and indication of both proton and carbon ion therapies (please add). - Response: There are presently only 6 centers worldwide having access to both, Proton- and Carbon Ion Radiotherapy. Having said that, we believe reporting on carbon-ion treatments in these rare tumors is important. According to the reviewer’s suggestion we have added a table with potential pros and cons of PBT vs. CIRT in the DISCUSSION section. 8. The study could be greatly improved if the authors would add a clinical algorithm to guide clinicians in their routine practice when facing the radiotherapy of such patients. - Response: This was also added.